# Immobilization of Phospholipase D on Fe_3_O_4_@SiO_2_-Graphene Oxide Nanocomposites: A Strategy to Improve Catalytic Stability and Reusability in the Efficient Production of Phosphatidylserine

**DOI:** 10.3390/molecules30040912

**Published:** 2025-02-16

**Authors:** Huiyi Shang, Juntan Wang, Bishan Guo, Haihua Zhu, Huijuan Li

**Affiliations:** 1Institute of Chemistry, Henan Academy of Sciences, 266-38 Mingli Road, Zhengzhou 450046, China; shanghuiyi@hnas.ac.cn; 2Institute of Business Scientific, Henan Academy of Sciences, 87 Wenhua Road, Zhengzhou 450002, China; wangjuntan@hnas.ac.cn (J.W.); guobishan@hnas.ac.cn (B.G.); 3School of Veterinary Medicine, Henan University of Animal Husbandry and Economy, 6 Longzihu North Road, Zhengzhou 450046, China

**Keywords:** phospholipase D, immobilization, phosphatidylserine, magnetite nanoparticle, graphene oxide

## Abstract

Phospholipase D (PLD) plays a pivotal role in the biosynthesis of phosphatidylserine (PS), but its practical application is constrained by limitations in stability and reusability. In this study, we successfully fabricated the Fe_3_O_4_@SiO_2_–graphene oxide (GO) nanocomposite by chemical binding of Fe_3_O_4_@SiO_2_ and GO. Subsequently, PLD was immobilized onto the nanocomposite via physical adsorption, with the aim of enhancing catalytic stability, reducing mass transfer resistance, and improving reusability. Under optimal conditions, the immobilization efficiency reached 84.4%, with a PLD loading capacity of 111.4 mg/g_support_. The optimal pH for PS production by immobilized PLD shifted from 6.0 to 6.5, while the optimal temperature increased from 45 °C to 50 °C. Notably, the immobilized PLD demonstrated a shorter reaction time and a higher PS yield, achieving a 95.4% yield within 90 min, compared to the free PLD (78.1% yield within 150 min), representing a 1.04-fold improvement in production efficiency. Furthermore, the immobilized PLD exhibited outstanding storage stability and thermal stability, along with remarkable reusability. Even after being reused for 10 cycles, the PS yield still stays as high as 78.3%. These findings strongly suggest that the Fe_3_O_4_@SiO_2_–GO immobilized PLD has the potential for the efficient production of PS.

## 1. Introduction

Phosphatidylserine (PS) is a unique functional phospholipid that plays a crucial role in cell signaling and numerous physiological processes [1,2]. Consequently, it has widespread applications in the functional food and pharmaceutical sectors. Dietary supplementation with PS can be instrumental in preventing Alzheimer’s disease, ameliorating memory, heightening vigilance and attention, alleviating depression, and mitigating stress [3,4,5,6,7]. Additionally, PS can function as an efficacious sports nutrition supplement to prevent post-exercise physical deterioration and expedite recovery [8]. Furthermore, PS can be utilized as a fundamental component of liposomes for the treatment of diseases such as hypercholesterolemia, ulcerative colitis, and type 1 diabetes [9,10,11]. Nevertheless, the natural occurrence of PS is rather scarce, and direct extraction incurs a relatively high cost. Hence, the transphosphatidylation reaction mediated by phospholipase D (PLD) has gained prominence. This method utilizes the highly abundant phosphatidylcholine (PC) as substrates to produce PS, offering mild reaction conditions, high conversion rates, and excellent selectivity [12].

Over recent decades, numerous PLDs have been identified in plants, animals, and microorganisms [13]. The activity of PLD can exhibit selectivity, favoring either hydrolysis or transphosphatidylation, depending on the source of the enzyme. Notably, Streptomyces PLDs have garnered significant attention due to their enhanced phosphatidyl transfer activity, broader substrate specificity, and more straightforward preparation process, making them the preferred catalyst for PS synthesis [14]. Typically, the transphosphatidylation reaction occurs in a biphasic system comprising water and an organic solvent that is immiscible with water [15]. However, prolonged exposure to organic solvents negatively impacts the structural integrity of PLD. Additionally, significant mass transfer resistance within the biphasic system can impede the efficiency of PS synthesis. Furthermore, challenges related to the storage stability and reusability of free enzymes contribute to elevated production costs, severely restricting the industrial application of PLD. To mitigate these constraints, researchers have explored various strategies, including displaying PLD on microbial cell surfaces to fabricate whole-cell catalysts [16,17], employing a nonaqueous or purely aqueous system [18,19,20], and immobilizing PLD [21,22,23]. Among these approaches, immobilization technology is widely adopted owing to its versatility in raw material selection, simplicity in process design, and ease of recovery.

To date, multiple immobilization techniques have been utilized to affix PLD onto various supports with the aim of fabricating immobilized PLDs. Certain investigators have opted for physical adsorption as a means to immobilize PLD onto appropriate supports, exemplified by macroporous resins [23], macroporous SiO_2_/cationic polymers [24], novel epoxy resin hierarchical porous polymers [25], and lignin nanoparticles [26]. Zhang et al. achieved the immobilization of PLD on amino-functionalized hollow mesoporous silica cubes through glutaraldehyde (GA) cross-linking [27]. Zhao et al. realized the immobilization of PLD by integrating the adsorption-precipitation-cross-linking method with bio-imprinting [28]. Some researchers have attained the immobilization of PLD by means of enzyme–inorganic/metal hybrid nanoflower technology [29,30]. Additionally, some have successfully anchored the PLD fusion protein on cellulose materials by constructing a fusion protein of PLD and cellulose-binding domain [31,32]. Owing to these extensive efforts, the catalytic efficiency, stability, and reusability of the immobilized PLD have all been ameliorated. Nevertheless, it must be underlined that in industrial settings where simplicity and cost-efficiency are of utmost importance, no one-size-fits-all approach exists applicable to any given enzyme. Hence, it is imperative to explore and implement novel materials and methods in the immobilization of PLD.

Magnetic graphene oxide (GO) has garnered widespread interest in enzyme engineering due to its superparamagnetism, two-dimensional architecture, large specific surface area, substantial enzyme loading capacity, and easy recyclability [33,34,35,36]. In many reports, GO magnetic composites are typically synthesized through simple co-precipitation or electrostatic adsorption methods. Nevertheless, when these composites are employed as immobilization supports, several issues come to the fore. Firstly, the unprotected Fe_3_O_4_ is highly unstable, given its propensity to be oxidized into other substances in an aerobic environment, thereby diminishing the magnetic and dispersive properties of the material [37,38,39]. Additionally, the chemically active surface of the exposed Fe_3_O_4_ can trigger the inactivation of enzymes. Secondly, most adsorption sites on the GO surface are occupied by magnetic particles, rendering them inaccessible for further enzyme binding. Finally, composites formed through simple physical adsorption may disintegrate over long-term usage, leading to the leaching out of Fe_3_O_4_ particles and reduced material recovery performance. To address these issues, magnetic particles can be encapsulated with appropriate materials, such as silica or polymer, to form a core/shell structure [40,41,42], which is then covalently bonded to GO. This approach not only augments the dispersibility of the material but also bolsters its stability [43].

In our previous work, we fabricated magnetic graphene oxide via the impregnation method and achieved the immobilization of PLD through GA after amino modification, with a PS yield reaching 92.8% [44]. Hence, in this work, we aim to develop a covalently linked Fe_3_O_4_@SiO_2_–GO nanocomposite, serving as an efficient support with enhanced dispersibility and easy separability. Subsequently, the immobilization of PLD is to be realized through physical adsorption, with the anticipation of further augmenting the performance of the immobilized enzyme as well as the PS yield.

## 2. Results and Discussion

### 2.1. Characterization of Fe_3_O_4_@SiO_2_–GO Nanoparticles

#### 2.1.1. FTIR Analysis

The FTIR spectra of GO, Fe_3_O_4_@SiO_2_, and Fe_3_O_4_@SiO_2_–GO are shown in Figure 1. All three samples exhibit a broad peak around 3400 cm^−1^. This peak is attributed to the stretching vibration of O–H bonds caused by surface hydroxylation and adsorbed water molecules [45]. For the GO sample, the peaks observed at 1735 cm^−1^, 1618 cm^−1^, and 1050 cm^−1^ are assigned to the C=O stretching, C=C aromatic bending, and C–O–C stretching vibrations of alkoxy groups, respectively (Figure 1a) [46]. In Figure 1b, the peak observed at 1630 cm^−1^ is potentially associated with the bending vibration of water molecules adhered to the nanoparticles. The peak at 1100 cm^−1^ represents the characteristic stretching vibration of the Si–O bond within the silica material, while the peak at 572 cm^−1^ stems from the stretching vibration of the Fe–O bond [40]. This substantiates the successful coating of the Fe_3_O_4_ nanoparticles with silica on their surfaces. In Figure 1c, in addition to the 1091 cm^−1,^ which pertains to the stretching vibration of the Si–O bond, the 1050 cm^−1^ corresponding to the stretching vibration of the C–O bond, and the 585 cm^−1^ attributable to the stretching vibration of the Fe–O bond. The original peak at 1735 cm^−1^ in GO has nearly vanished, the newly emerged peak at 1638 cm^−1^ is likely to correspond to the C=O stretching of the amide bond, and the peak at 1380 cm^−1^ is associated with the C–N stretching vibration and N–H bending vibration of the amide bond. Moreover, the characteristic peak manifesting at 2975 cm^−1^ is ascribed to the stretching vibration of –CH–, which can be traced back to the C–H stretching of 3-aminopropyltriethoxysilane (APTES) [47,48]. These findings suggest that the APTES-functionalized Fe_3_O_4_@SiO_2_ has been covalently conjugated to GO via the amide bond. Finally, the adsorption of PLD onto Fe_3_O_4_@SiO_2_–GO was confirmed by the enhanced intensity of the amide I band at ~1640 cm^−1^ and the emergence of the characteristic amide II band at ~1520 cm^−1^ in the FTIR spectra (Figure 1d).

#### 2.1.2. Morphology

The morphology and architecture of the synthesized nanoparticles were investigated by scanning electron microscope (SEM). In Figure 2a, the SEM micrograph of the GO sample is depicted. GO manifests a lamellar structure with a multitude of corrugations on its surface. Figure 2b illustrates the SEM image of the Fe_3_O_4_ nanomaterial. Ordinarily, the Fe_3_O_4_ particles assume a spherical shape, with diameters spanning from 20 to 70 nm and an average dimension of roughly 40 nm. Subsequent to being encapsulated by silica (Figure 2c), the average particle size of the newly generated Fe_3_O_4_@SiO_2_ augments to 190 nm, and the dispersity is conspicuously augmented. When covalently conjugated with GO following amino functionalization, the novel Fe_3_O_4_@SiO_2_–GO nanocomposite exhibits a coarse outer surface and thickens (Figure 2d). This is attributable to the amino-functionalized magnetic nanosilica material adhering to the surface of the GO sheets via the formation of amide bonds. The existence of amide bonds guarantees the stability of the Fe_3_O_4_@SiO_2_–GO nanocomposite, impeding the facile leakage of magnetic particles from the material.

#### 2.1.3. VSM Measurements

In order to assess the magnetic properties of the synthesized nanoparticles, the magnetic hysteresis loops of the samples were measured using a vibrating sample magnetometer within the magnetic field range spanning from −15,000 to 15,000 Oe at room temperature (Figure 3). The saturation magnetization values of Fe_3_O_4_@SiO_2_ and Fe_3_O_4_@SiO_2_–GO were 27.5 emu/g and 21.3 emu/g, respectively. Upon the introduction of GO, the magnetization intensity of the magnetic nanocomposite witnessed a decline, which was in line with the previous reports documented in the literature [45,46]. Despite the reduction in the saturation magnetization, the magnetic nanocomposite remained capable of being completely separated from the solution under the influence of an external magnetic field.

### 2.2. The PLD Immobilization Parameters

During the immobilization process, factors such as initial PLD volume, pH, temperature, and contact time were examined for their impacts on immobilization efficiency and the activity of the immobilized enzyme.

#### 2.2.1. The Initial PLD Volume

Under the conditions of 25 °C, pH 7.0, and a contact time of 180 min, the influence of different amounts of PLD solution added on the immobilization efficiency and enzyme activity was investigated. As illustrated in Figure 4a, the immobilization efficiency decreased as the initial PLD volume increased, while the relative activity of the immobilized catalyst exhibited a trend of initially increasing and subsequently decreasing. Similar phenomena have been reported in several studies [27,49]. This is because when the amount of enzyme is low, a smaller amount of enzyme is adsorbed onto the carrier, resulting in lower activity of the immobilized catalyst. As the initial PLD volume increases, the loading capacity of the carrier reaches saturation, and the excess enzyme cannot be adsorbed, thereby reducing the immobilization rate. However, when the enzyme dosage exceeds 0.4 mL, the activity of the immobilized enzyme shows a downward trend. This might be due to the fact that the PLD loading on the carrier is too high, causing excessive enzyme molecules to stack into an enzyme shell, generating severe mass transfer resistance, and ultimately leading to a decrease in the activity of the immobilized PLD. Therefore, 0.4 mL of the PLD solution was selected for immobilization, at which point the enzyme/support ratio was 132 mg_protein_/g_support_, the immobilization efficiency was 74.6%, and the loading capacity of the magnetic nano-composite support was 98.5 mg/g_support_.

#### 2.2.2. pH

The pH of the solution plays a crucial role in both enzyme activity and the adsorption process [50]. It affects the morphology and surface charge of the immobilization support as well as that of the enzyme. In this study, we investigated the impact of solution pH on immobilization efficiency and the activity of the immobilized enzyme within the pH range of 4.0 to 9.0. As illustrated in Figure 4b, within the pH interval of 4.0 to 6.0, both immobilization efficiency and enzyme activity augment with increasing pH. Conversely, beyond pH 6.0, with a continued rise in pH, both immobilization efficiency and enzyme activity display a downward trend. Hence, a pH of 6.0 was designated as the optimal pH, at which the immobilization efficiency attained 81.6%. According to the literature, the point zero charge (pHpzc) of Fe_3_O_4_@SiO_2_–GO lies between 5.7 and 6.2 [51,52]. Below this pHpzc, the support’s surface carries a positive charge; above it, a negative charge. PLD has a theoretical isoelectric point (pI) of 5.9, closely aligned with the pHpzc of the support. This results in electrostatic repulsion when either pH is less than or greater than 6.0 due to both surfaces having the same type of charge. These repulsions are minimized only at a pH near 6.0. Moreover, near its pI, the solubility of PLD decreases, enhancing its propensity to precipitate and facilitating its adsorption onto the nanocomposite support.

#### 2.2.3. Temperature

Subsequently, the temperature was varied from 15 to 45 °C to investigate its effects on immobilization efficiency and the activity of the immobilized catalyst, as illustrated in Figure 4c. Within the 15–30 °C, both parameters exhibited gentle responses to temperature variation: immobilization efficiency increased from 74.1% to 84.1%, while relative activity rose from 88.2% to 100%. Conversely, in the 30–45 °C range, immobilization efficiency decreased from 84.1% to 75.6%, and relative activity sharply dropped to 75.8%. This can be attributed to the fact that an appropriate elevation in temperature enhanced the surface activity of the support, thereby augmenting its adsorption capacity for the PLD [53]. Nevertheless, a further increase in temperature led to aggregation among enzyme molecules, which, conversely, weakened the interaction between the support and the enzyme, resulting in desorption [54]. Additionally, high temperatures easily disrupt hydrogen and ionic bonds within enzyme molecules, altering their spatial conformation and reducing their catalytic performance [55]. Consequently, the decrease in activity at higher temperatures is more significant than the reduction in immobilization efficiency. Based on these findings, 30 °C was chosen as the optimal immobilization temperature.

#### 2.2.4. Contact Time

To ascertain the impact of the contact time between the support and the enzyme solution on PLD immobilization, the immobilization efficiency and relative activity of the immobilized catalyst were tested over different time durations. As depicted in Figure 4d, the adsorption of PLD onto the nanocomposite support was expeditious and progressively intensified. Within 60 min, the immobilization efficiency attained 75.9%, and by 90 min, it reached 84.4%. Subsequently, with the further elongation of the contact time, the immobilization efficiency ceased to augment. This phenomenon can be ascribed to the fact that, in the nascent stage of contact, copious vacant active sites were present on the support surface, which could be utilized for adsorption, thereby precipitating a swift escalation in the immobilization efficiency. As the contact time persisted beyond 90 min, the accessibility of unoccupied surface loci diminished, culminating in the adsorption efficiency reaching a plateau. At this juncture, the adsorption process achieved an equilibrium state, and concurrently, the relative activity of the immobilized enzyme under such circumstances was maximized. Hence, 90 min was designated as the optimal immobilization time, achieving an adsorption capacity of 111.4 mg/g_support_.

### 2.3. Optimization of the Transphosphatidylation Reaction

#### 2.3.1. Reaction Time

The investigation of reaction time in biocatalytic processes is essential for identifying the optimal duration required to achieve maximum product yield, thus enhancing economic viability. Figure 5a illustrates the progression of PS production under the catalysis of both free and immobilized PLD over time. During the initial stages, PS yields escalate rapidly under both catalyst systems. At 60 min, the yield for the free PLD attains 59.7%, whereas that for the immobilized PLD peaks at 79.6%, marking a 33.6% relative increase. Following this, the immobilized PLD’s yield maintains its steep rise, culminating in a peak of 95.4% at 90 min. Conversely, the free PLD’s yield growth decelerates, reaching its zenith of 78.1% at 150 min. The production efficiency (PS yield per unit time) of immobilized PLD can be affected by factors such as enzyme source, immobilization methods, and reaction conditions. Li et al. reported that immobilizing PLD on a ZnO nanowires/macroporous SiO_2_ composite support through an in situ cross-linking method significantly increased PS yield from 68.2% (within 60 min) to 94.8% (within 40 min), representing a 1.08-fold increase in production efficiency [22]. Zhang et al. reported that immobilization of PLD derived from *Streptomyces chromofuscus* on amino-modified hollow mesoporous silica cubes via a chemical cross-linking method significantly enhanced the catalytic efficiency. The PS yield improved from 76.3% within 12 h to 90.4% within 10 h, indicating a 30% increase in production efficiency [27]. In this study, the immobilized PLD achieved a maximum PS yield of 95.1% within 90 min, representing a 1.04-fold increase in production efficiency compared to free PLD.

#### 2.3.2. Effect of Substrate Concentration Ratio

Figure 5b displays the PS yield from free and immobilized PLD at varying mass ratios of L-serine to PC. Both systems demonstrate similar trends, but across all tested ranges, the PS yield generated by the immobilized PLD catalysis is consistently higher than that produced by the free PLD. At a substrate mass ratio of less than 3, the PS yield rises sharply as the proportion of L-serine increases; beyond this threshold, further increments in the L-serine proportion yield minimal improvement in PS yield. Therefore, a substrate mass ratio of 3 was regarded as the optimum. Under this condition, the PS yield of the immobilized PLD was 94.9%, while that of the free PLD was 78.2%.

The transphosphatidylation reaction of PLD typically occurs in a biphasic system comprising water and an organic solvent that is immiscible with water, wherein the phospholipid substrate resides in the organic phase, and the enzyme and nucleophilic substrate are present only in the aqueous phase [28,56]. This reaction unfolds at the interface between these phases, beginning with the nucleophilic attack of PLD on the first substrate PC to form a covalently linked phosphatidyl-enzyme intermediate. In the subsequent step, hydrolysis or transphosphorylation of the intermediate occurs via the second substrate (water molecule or alcohol). When L-serine acts as the second substrate, PS is synthesized; when water predominates, phosphatidic acid (PA) forms [14]. While several researchers opt for microaqueous or anhydrous systems to mitigate hydrolysis reactions [19,57], such approaches result in reduced reaction efficiency due to enzyme conformational changes. Therefore, the biphasic system continues to be a competitive option for industrial PS production. In the biphasic system, appropriately increasing the concentration of L-serine is conducive to improving the yield of PS.

#### 2.3.3. pH

In this study, we investigated the PS yield generated by both free and immobilized PLD over a pH range of 4.0 to 9.0. The results are illustrated in Figure 5c. For the free PLD, the highest PS yield was observed at pH 6.0, reaching 78.2%. As the pH continued to increase, the PS yield rapidly decreased, dropping to only 27.7% at pH 9.0. In contrast, the immobilized PLD reached its highest PS yield at pH 6.5, with a yield of 95.1%, which is 1.21 times higher than that of the free PLD. Furthermore, the PS yield for the immobilized PLD varied more smoothly with changes in pH, retaining 74.2% PS yield at pH 9.0, almost equivalent to the optimal PS yield of the free PLD. This is because the microenvironment provided by the nanocomposite material in the immobilized PLD, along with the interaction between the enzyme and the material, helps to reduce the sensitivity of the enzyme molecules to changes in environmental pH, thereby broadening the range of applications for the catalyst.

#### 2.3.4. Temperature

The reaction temperature is a critical parameter affecting enzymatic activity, stability, and substance diffusion in the reaction system. Consequently, the PS yield of both immobilized and free PLD was systematically investigated across different temperatures. As depicted in Figure 5d, within the temperature range of 30 to 60 °C, both PLD forms exhibited an initial increase followed by a decline in PS yield. Specifically, immobilized PLD reached its maximum PS yield at 50 °C, amounting to 94.8%, whereas the free PLD’s peak yield was observed at 45 °C, with a yield of 79.2%. As the temperature continued to rise to 60 °C, the PS yield of the free PLD dropped to 54.1%, which is only 68.3% of its optimum yield. In contrast, the immobilized PLD maintained an 85.7% PS yield at 60 °C, accounting for 90.3% of its optimal level. This behavior can be attributed to the increased energy of both enzyme and substrate with rising temperatures, enhancing the collision frequency between enzyme and substrate and improving mass transfer efficiency. These factors collectively boost catalytic efficiency. However, excessively high temperatures may disrupt the enzyme structure, leading to decreased catalytic activity and lower PS yield. Immobilized PLD exhibits enhanced structural stability due to interactions between the support and enzyme. This increased stability allows it to withstand higher temperatures, shifting its optimum temperature range and enabling the maintenance of a high PS yield over a broader temperature range.

### 2.4. Stability and Reusability

#### 2.4.1. Thermal Stability

Thermal stability is a critical parameter for the large-scale production and industrial application of immobilized biocatalysts. To evaluate the thermal stability of immobilized PLD, the enzyme was incubated at temperatures ranging from 25 to 65 °C for 1 h and compared with free PLD. As illustrated in Figure 6, immobilized PLD retained 93.4% of its initial activity after incubation at 50 °C for 1 h, significantly higher than the 65.2% retained by free PLD. At 65 °C for 1 h, free PLD activity decreased to 10.1%, whereas immobilized PLD maintained 79.4% of its original activity. This clearly demonstrates that the thermal stability of immobilized PLD is considerably superior to that of free PLD. Immobilized enzymes generally exhibit greater thermal resistance compared to their free-form, which is attributed to the restriction of enzyme mobility and increased rigidity [58]. Consequently, this improved thermal stability enhances the catalytic efficiency of the immobilized enzyme in PS synthesis as reactions proceed more rapidly at higher temperatures, leading to higher product yields [59].

#### 2.4.2. Storage Stability

To evaluate the storage stability, the activities of the free PLD and immobilized PLD were measured at different time points during storage at 4 °C, with the original activity recorded at 100%. Figure 7 demonstrates the changes in relative activity of two forms of PLD during storage. It is evident that the decline in activity of free PLD is significantly faster than that of immobilized PLD. Upon reaching a 40-day storage duration, the free PLD retained just 50% of its initial activity, contrasting the immobilized PLD, which sustained 85% of its beginning activity. Further prolonging the storage to 50 days revealed that the immobilized PLD still retained 82% of its initial activity, while the free PLD only maintained a survival of 44%. In summary, when considering long-term stability, the immobilized PLD demonstrates a significant advantage over the free form, making it more suitable for industrial uses.

#### 2.4.3. Reusability of Immobilized PLD

Immobilized enzymes, characterized by their ease of separation and recyclability, offer significant advantages in industrial production. In this study, we constructed an immobilized PLD incorporating magnetic nanoparticles, thereby endowing the enzyme with magnetic responsiveness. Under an external magnetic field, the immobilized PLD can be swiftly separated from the reaction system for use in new batches of production, substantially reducing enzyme usage costs. To assess the reusability of the immobilized PLD, multiple rounds of PS synthesis were conducted using the same batch of immobilized enzymes, with PS yield quantified after each cycle. As shown in Figure 8, the PS yield was maintained at 85.1% after five cycles of repeated use. Although the immobilization support protects the enzyme during reactions, continuous cycling alters the microenvironment of PLD, resulting in partial denaturation. Furthermore, vigorous stirring can cause some enzymes to detach from the support, decreasing subsequent reaction yields. Nonetheless, after 10 consecutive reactions, the PS yield of our immobilized enzyme remained at a commendable 78.3%. This stability is attributed to the excellent mechanical properties of the nanocomposite material and favorable interactions between the enzyme and the nanocomposite, thereby enhancing the conformational stability of PLD. These results demonstrate that Fe_3_O_4_@SiO_2_–GO–PLD exhibits excellent reusability and is suitable for the industrial production of rare phospholipids such as PS.

### 2.5. Comparison of PLD Immobilization on Fe_3_O_4_@SiO_2_–GO and Other Supports

In the literature, PLDs have been immobilized on various supports via a variety of methods, as summarized in Table 1. These approaches have collectively enhanced enzyme stability, facilitated enzyme recycling, and improved PS yield. Our work distinguishes itself in several key aspects. First, the selection of magnetic materials enables rapid recovery and reuse via an external magnetic field, offering a more energy-efficient alternative to centrifugation, which is particularly advantageous for industrial applications. Second, compared to our previous research, the material developed in this study exhibits a significant increase in loading capacity (from 90.3 to 111.4 mg/g_support_) and immobilization efficiency (from 30.1% to 84.4%). This improvement can be attributed to two factors: (1) covalent bonding with GO rather than simple co-precipitation, which introduces more GO and increases the specific surface area, thereby enhancing loading capacity, and (2) coating Fe_3_O_4_ with silica significantly improves the dispersibility and stability of the material. In comparison to other studies listed, the enzyme loading and immobilization efficiency of our immobilized PLD, while not the highest, remain competitive. Third, we utilized physical adsorption for immobilization, a method that is simpler than cross-linking or covalent binding and effectively preserves enzyme activity. Finally, the immobilized PLD we developed exhibits strong performance in both PS yield and reaction time, demonstrating its competitiveness in these key metrics. In conclusion, we have successfully developed a high-performance immobilized catalyst that demonstrates substantial potential for application in the industrial-scale production of PS.

## 3. Materials and Methods

### 3.1. Materials

Tetraethyl orthosilicate (TEOS) and the standard sample of choline chloride were procured from Aladdin Biochemical Technology Co., Ltd. (Shanghai, China). 1-ethyl-3-(3-dimethylaminopropyl) carbodiimide (EDC) and N-hydroxy-Succinimide (NHS) were obtained from Macklin Biochemical Technology Co., Ltd. (Shanghai, China). Choline oxidase and peroxidase were sourced from Sigma-Aldrich Trading Co., Ltd. (Shanghai, China). For other materials, including PLD, PC, and L-serine, references can be made to our previous papers [41].

### 3.2. Preparation of Fe_3_O_4_@SiO_2_–GO

Fe_3_O_4_@SiO_2_ core/shell nanoparticles were synthesized using the following procedures. First, Fe_3_O_4_ nanoparticles were dispersed in a solution consisting of 50 mL of water and 100 mL of isopropyl alcohol. The pH was adjusted to 11 using a concentrated ammonia solution. Then, 2.0 mL of TEOS was added dropwise while stirring gently at low speed, allowing the reaction to proceed for 12 h at 25 °C. The product was collected by magnetic separation, washed, and dried. To obtain amino-functionalized Fe_3_O_4_@SiO_2_, 100 mg of the as-prepared Fe_3_O_4_@SiO_2_ was dispersed in a mixture of 50 mL anhydrous ethanol and 5 mL water via ultrasonication. Following this, 0.2 mL of 3-aminopropyltriethoxysilane (APTES) was added, and the mixture was stirred for 12 h, washed thoroughly, and dried.

For the synthesis of Fe_3_O_4_@SiO_2_–GO nanocomposite, GO was first prepared from graphite powder using a modified Hummers’ method [60]. Subsequently, 100 mg of GO was dispersed in 300 mL of water via ultrasonication for 3 h. Then, 50 mg EDC and 40 mg NHS were added, and the mixture was stirred for 30 min and further sonicated for 1 h to create a homogeneous suspension. Next, 100 mg of the amino-functionalized Fe_3_O_4_@SiO_2_ was incorporated into the suspension, followed by additional sonication for 1 h. Finally, the mixture was reacted with constant stirring at 80 °C for 1 h. The resulting Fe_3_O_4_@SiO_2_–GO nanocomposite was harvested via magnetic separation, washed with distilled water, and dried. A complete scheme of nanocomposite is illustrated in Figure 9.

### 3.3. Characterization Methods

The morphology and microstructure of the samples were characterized using a scanning electron microscope (SEM; SU8010, Hitachi, Tokyo, Japan) operated at an accelerating voltage of 5 kV. The chemical composition of the samples was analyzed via Fourier transform infrared spectroscopy (FTIR; Nicolet 6700, Thermo Fisher Scientific, Waltham, MA, USA), with spectra recorded between 400 and 4000 cm^−1^. The magnetic properties of the samples were measured at room temperature using a vibrating sample magnetometer (VSM, 3107, East Changing, Beijing, China).

### 3.4. Immobilization of PLD

In this study, the physical adsorption method was employed to immobilize PLD onto the magnetic nanocomposite. Specifically, 4 mg of the as-prepared Fe_3_O_4_@SiO_2_–GO nanocomposite was first dispersed in 1.4 mL of phosphate buffer solution (0.1 M, pH 7.0). Then, 0.4 mL of a diluted PLD enzyme solution (1.32 mg/mL, pH 7.0) was introduced, and further phosphate buffer (0.1 M, pH 7.0) was added to bring the total volume of the system up to 2 mL. Subsequently, the mixture was placed in a shaker at 25 °C and agitated at 220 rpm for 180 min to allow for adsorption. Afterward, the nanomaterial was separated using an external magnetic field, washed with phosphate buffer, freeze-dried, and stored at 4 °C for future use. The final product was designated as Fe_3_O_4_@SiO_2_–GO–PLD.

To optimize the immobilization efficiency and the activity of the immobilized enzyme, key parameters were systematically investigated, including the initial volume of the PLD solution (0.1–0.6 mL), pH (4.0–9.0), reaction temperature (15–45 °C), and contact time (20–240 min).

The immobilization efficiency can be computed by using the following Equation (1):(1)$$\mathrm{Immobilization}\;\mathrm{efficiency}\;\left(\%\right)=\left(A_{0}-A_{1}\right)/A_{0}\times 100\%$$
where A_0_ and A_1_ represent the total amount of PLD introduced into the solution and the number of unbound PLD, respectively. The concentration of PLD in the supernatant was determined through the use of a BCA Protein Assay Kit (TaKaRa Biotechnology (Beijing) Co., Ltd., Beijing, China).

### 3.5. Enzyme Activity Assay

The PLD activity in this study, which was represented as a hydrolytic activity because of its convenient determination, was measured by employing the method developed by Imamura and Horiuti with certain modifications [61]. The reaction mixture consisted of 200 μL of 10 mg/mL PC, 100 μL of 0.2 M Tris-HCl (pH 5.5), 100 μL of 50 mM calcium chloride, 50 μL of 1% Triton X-100, and 50 μL of the enzyme solution (or 1 mg immobilized PLD). After incubation at 37 °C for 10 min, the reaction was terminated by adding 50 µL of a solution containing 50 mM EDTA and 100 mM Tris-HCl (pH 8.0). The resulting mixture was combined with 3 mL of a solution containing 2 U of choline oxidase, 2 U of horseradish peroxidase, 0.3 mg of 4-aminoantipyrine, 1 mg of phenol, and 100 mM Tris-HCl (pH 8.0). After reacting at 37 °C for 30 min, the absorbance of the reaction mixture was measured at 500 nm using a microplate reader. A calibration curve was obtained by using a standard choline chloride solution in place of the enzyme reaction solution. The unit of the hydrolytic activity of PLD was defined as the amount of enzyme that generates 1 mM choline per minute. The relative activity was computed as following Equation (2):(2)$$Relative\;activity\left(\%\right)=B_{1}/B_{0}\times 100\%$$
where B_1_ is the group’s enzyme activity, and B_0_ is the group’s highest enzyme activity.

### 3.6. Synthesis of PS

PS was synthesized using L-serine and PC as substrates in a butyl acetate aqueous solution system. Specifically, PC was dissolved in 2 mL of butyl acetate at a concentration of 20 mg/mL, thereby serving as the organic phase. Meanwhile, L-serine was dissolved in 1 mL of acetate buffer (0.1 M, pH 6.0) at a concentration ranging from 20 to 120 mg/mL, acting as the aqueous phase. The additional amount of the immobilized PLD was 3 mg (dry weight), and that of the free PLD was 250 μL. The reaction was carried out at 45 °C with stirring at 220 rpm for 180 min. To maximize the PS yield, a series of experiments were conducted using varying process parameters, including reaction time, L-serine concentration, buffer pH, and reaction temperature. The PS yield and production efficiency were computed using the following equation:(3)$$PS\;yield\left(\%\right)=m_{ps}/m_{pc}\times 100\%$$(4)$$Production\;efficiency\left(\%\right)=PS\;yield/Reaction\;time$$

### 3.7. Thermal Stability

To evaluate and compare thermal stability, free PLD and immobilized PLD were incubated at different temperatures (25–65 °C) for 60 min without substrate. The residual activities of both free and immobilized PLD were subsequently measured, with the activity at 25 °C serving as the control and normalized to 100%.

### 3.8. Storage Stability and Reusability

The storage stability was assessed by monitoring the residual activities of both free PLD and immobilized PLD at various time points during storage at 4 °C.

The reusability of the immobilized PLD was evaluated by quantifying the PS yield during successive cycles of reuse. After each batch of reaction, the immobilized PLD was recovered through magnetic separation, subsequently washed with phosphate buffer (0.1 M, pH 7.0), and then utilized for the subsequent batch reaction. The immobilized PLD can be repeatedly employed for multiple rounds of PS production by adding fresh substrates under optimal conditions.

### 3.9. Statistical Analysis

The experiment was carried out in triplicate. Statistical analysis was performed by utilizing SPSS 25.0 and Origin 2024 software. The results are presented as mean ± standard deviation.

## 4. Conclusions

Fe_3_O_4_@SiO_2_–GO magnetic nanoparticles were synthesized via covalent bonding and subsequently utilized as a support for the immobilization of PLD through physical adsorption. The resulting material demonstrated outstanding magnetic separation capabilities and adsorption capacity. Upon the immobilization of PLD, the thermal and storage stabilities of the immobilized PLD were markedly enhanced. Furthermore, the PS yield of the immobilized PLD achieved 95.4% under optimized conditions (50 °C, pH 6.5, 90 min), representing a 22.2% increase over the free PLD yield of 78.1% (45 °C, pH 6.0, 150 min), with a corresponding 104% improvement in production efficiency. Notably, even after 10 successive reaction cycles, the PS yield could still be retained at 78.3%. These findings indicate that Fe_3_O_4_@SiO_2_–GO–PLD demonstrates robust stability and high efficiency in PS production, suggesting its potential applicability for large-scale industrial manufacturing of PS.

## Figures and Tables

**Figure 1 molecules-30-00912-f001:**
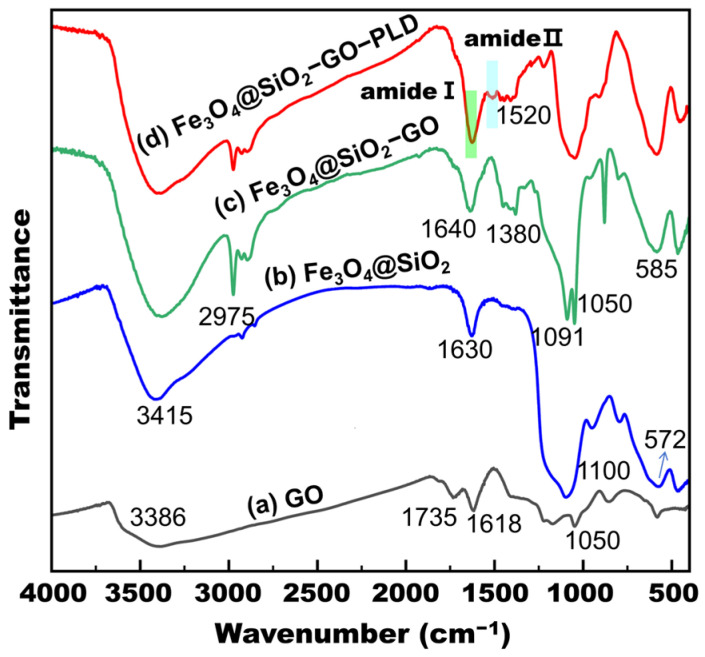
FTIR spectra of GO (**a**), Fe_3_O_4_@SiO_2_ (**b**), Fe_3_O_4_@SiO_2_–GO (**c**), and Fe_3_O_4_@SiO_2_–GO–PLD (**d**).

**Figure 2 molecules-30-00912-f002:**
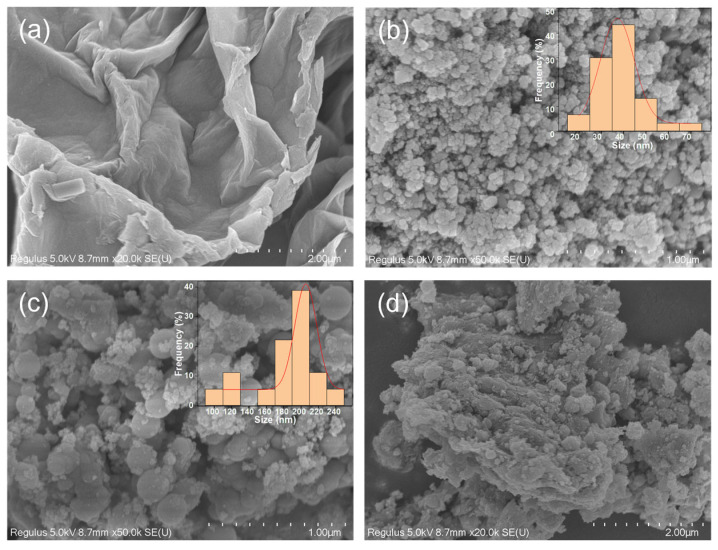
SEM images of GO (**a**), Fe_3_O_4_ (**b**), Fe_3_O_4_@SiO_2_ (**c**), and Fe_3_O_4_@SiO_2_–GO (**d**).

**Figure 3 molecules-30-00912-f003:**
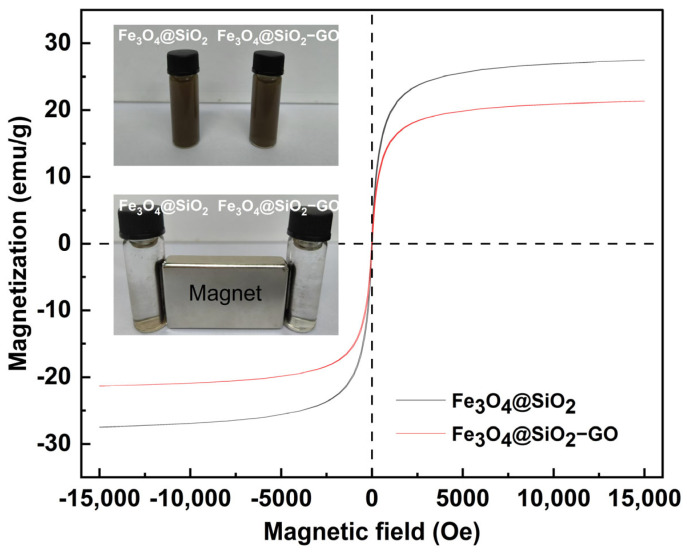
Magnetic hysteresis loops and the insets showing the Fe_3_O_4_@SiO_2_ and Fe_3_O_4_@SiO–GO in aqueous solution before and after magnetic separation by an external magnet.

**Figure 4 molecules-30-00912-f004:**
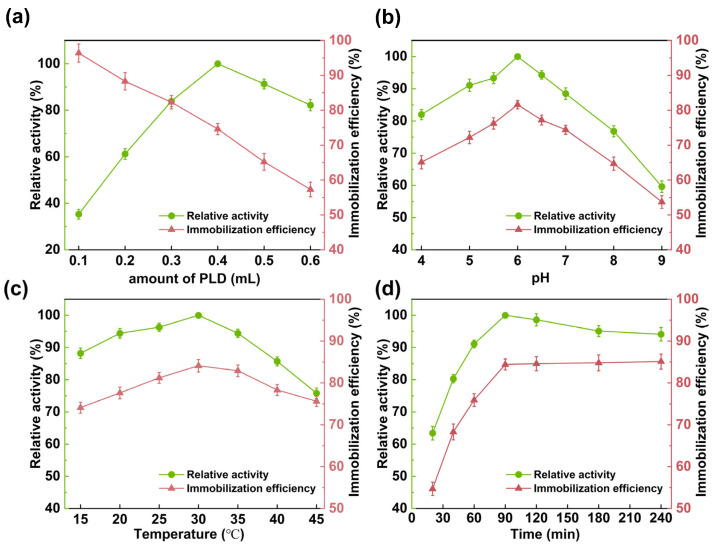
The effect of the initial PLD volume (**a**)**,** pH (**b**), temperature (**c**), and contact time (**d**) on the immobilization efficiency and the relative activity.

**Figure 5 molecules-30-00912-f005:**
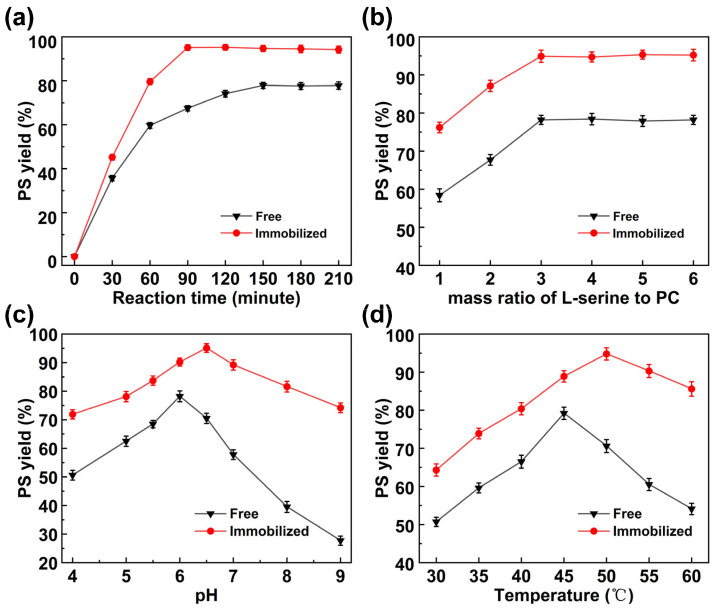
(**a**) Effect of reaction time on PS yield by free PLD (L-serine/PC 3/1, 45 °C, pH 6.0) and immobilized PLD (L-serine/PC 3/1, 50 °C, pH 6.5); (**b**) effect of mass ratio of L-serine and PC on PS yield by free PLD (45 °C, pH 6.0, 150 min) and immobilized PLD (50 °C, pH 6.5, 90 min); (**c**) effect of aqueous solution pH on PS yield by free PLD (L-serine/PC 3/1, 45 °C, 150 min) and immobilized PLD (L-serine/PC 3/1, 50 °C, 90 min); (**d**) effect of temperature on PS yield by free PLD (L-serine/PC 3/1, pH 6.0, 150 min) and immobilized PLD (L-serine/PC 3/1, pH 6.5, 90 min).

**Figure 6 molecules-30-00912-f006:**
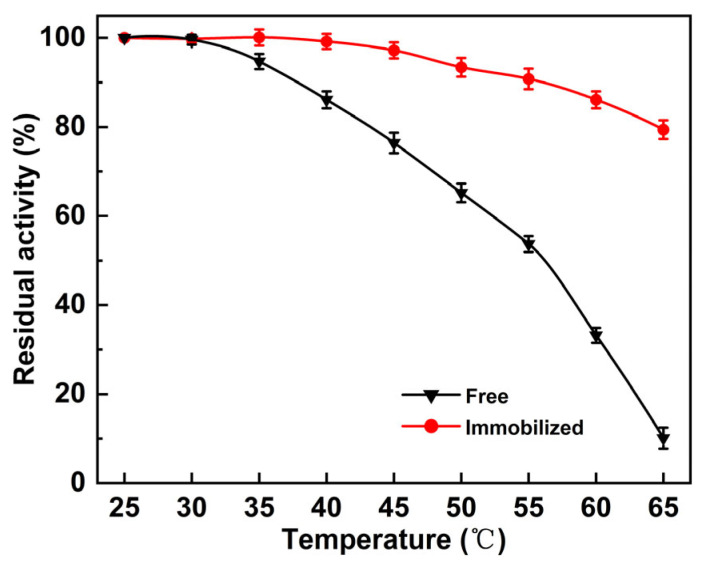
Dependence of residual activity on the temperature for 1 h incubation of the free and immobilized PLD.

**Figure 7 molecules-30-00912-f007:**
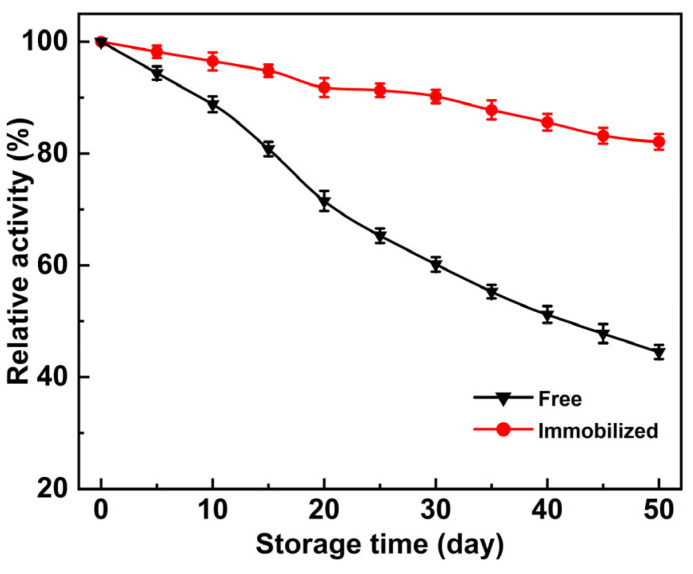
Evaluation of storage stabilities of the free and the immobilized PLD.

**Figure 8 molecules-30-00912-f008:**
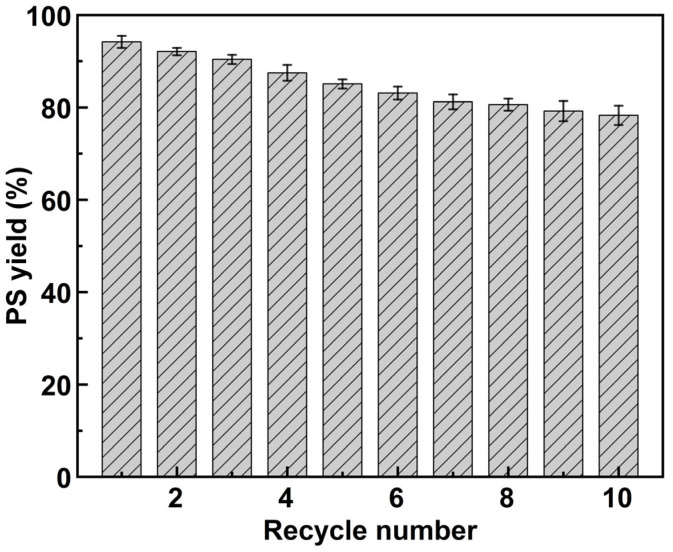
The catalytic reusability of Fe_3_O_4_@SiO_2_–GO–PLD.

**Figure 9 molecules-30-00912-f009:**
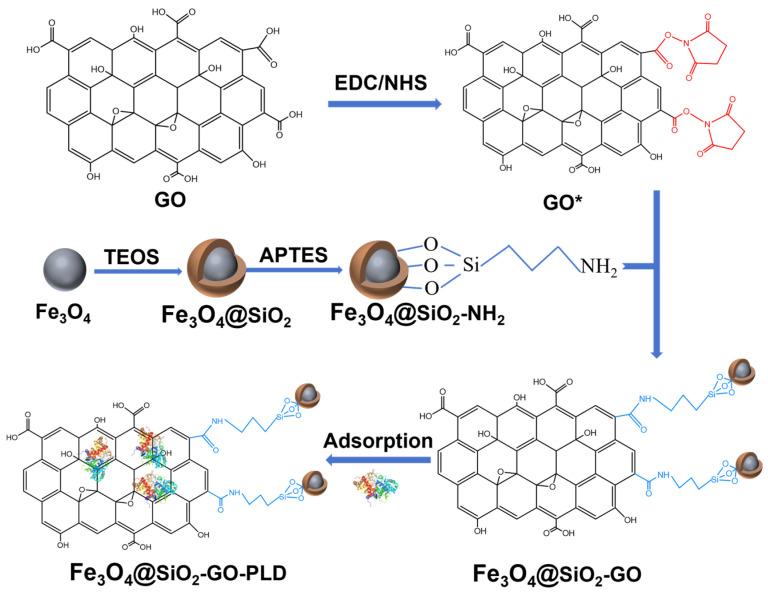
Scheme for the synthesis of Fe_3_O_4_@SiO_2_–GO and immobilization of PLD.

**Table 1 molecules-30-00912-t001:** The characteristic parameters of PLD are immobilized via diverse methods and supports.

Support/Method	Immobilization Efficiency (%)	Enzyme Loading (mg/g_support_)	PS Yield (%)	Time	Ref.
Fe_3_O_4_@SiO_2_–GO/adsorption	84.4	111.4	95.1	90 min	This study
magnetic GO/covalent linkage	30.1	90.3	92.8	6 h	[44]
non-porous SiO_2_/cross-linking	/	/	97	6 h	[28]
macroporous SiO_2_-cationic polymer/adsorption	/	61.5	96.2	40 min	[24]
epoxy resin hierarchical porous polymer/adsorption	/	223	95.5	40 min	[25]
cellulose nanofibrils/cellulose-binding domain	56.3	/	95.4	2 h	[31]
ZnO nanowires-macroporous SiO_2_/in situ cross-linking	/	68.1	94.8	40 min	[22]
ordered mesoporous silica cube/adsorption	76.27	/	91.2	2 h	[49]
amino hollow mesoporous silica cube/cross-linking	87.15	1.859	90.4	10 h	[27]

/ Data are not available in the references.

## Data Availability

All relevant data of this study are presented. Additional data will be provided upon request.

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
