# Peer review of "Immobilization of Phospholipase D on Fe_3_O_4_@SiO_2_-Graphene Oxide Nanocomposites: A Strategy to Improve Catalytic Stability and Reusability in the Efficient Production of Phosphatidylserine"

_molecules, 2025, doi:10.3390/molecules30040912_

Round 1
Reviewer 1 Report
Comments and Suggestions for Authors
In MS H. Shang et al. „Immobilization of Phospholipase D on Fe3O4@SiO2-Graphene Oxide nanocomposites: ... (ID: molecules-3453960)” Fe3O4@SiO2-GO immobilized Phospholipase D was fabricated, and basic characteristics of the complex is demonstrated.
The topic meets the scope of the journal, important for academic and industrial fields and should be interesting for possible readers of the journal.
I think the MS is written well, the structure of the MS is good, mistakes, misconceptions, invalid statements were not detected, the readability grammatically and literally is excellent – although, I have no qualification for evaluating the language.
I definitely support publishing it. I have only one minor comment and a couple of formal notes that the authors can take into consideration before finalizing the MS.
Comment:
I could not figure out why GO is used as support. The typical advantage of GO (special electric and mechanical properties) is not used here. If the large specific surface area is the question, there might be other systems (1-2-3D) for suitable support.
Formal notes:
- I prefer, usually, shorter titles of papers.
- There is no “Figure 1”, instead, there are two figures with numbering “Figure 2”.
- I would specify pH uniformly with one digit accuracy.
- P3,L93: “Finally,, composites ... „
- I prefer SI units in scientifc publication (e.g. Oe).
Final suggestion: accepting for publication in the present state (taking into possible consideration the above statements).
Author Response
Reviewer #1
Comments 1: The topic meets the scope of the journal, important for academic and industrial fields and should be interesting for possible readers of the journal. I think the MS is written well, the structure of the MS is good, mistakes, misconceptions, invalid statements were not detected, the readability grammatically and literally is excellent – although, I have no qualification for evaluating the language. I definitely support publishing it. I have only one minor comment and a couple of formal notes that the authors can take into consideration before finalizing the MS.
Response 1: We are extremely grateful for your overwhelmingly positive feedback on our manuscript.
Comments 2: I could not figure out why GO is used as support. The typical advantage of GO (special electric and mechanical properties) is not used here. If the large specific surface area is the question, there might be other systems (1-2-3D) for suitable support.
Response 2: Thank you very much for your insightful question regarding the use of GO as a support in our study. We truly appreciate your attention to this aspect, as it has provided us with an opportunity to clarify our reasoning.
While it is true that GO is well known for its remarkable electrical and mechanical properties, and these properties are not directly exploited in our work on immobilization, immobilization is indeed one of the important applications of GO. There is a substantial body of research demonstrating the effectiveness of GO as an immobilization support. In particular, numerous studies have focused on the immobilization of lipases using GO as the support material, and they have achieved promising results. Lipases and phospholipase D (PLD) share certain similarities in terms of their enzymatic characteristics and reaction mechanisms. Given these similarities, we hypothesized that GO could also serve as an effective support for the immobilization of PLD, which our experimental results have to some extent validated.
We are fully aware that there are various other materials in 1-2-3D structures that could potentially be used as immobilization supports. In fact, we have already planned to explore a wider range of materials in our future research. This will not only help us to comprehensively evaluate the performance of different supports but also enable us to further optimize the immobilization process of PLD.
Once again, thank you for your valuable comment. Your feedback has been instrumental in guiding our research, and we believe that with continuous exploration and improvement, our research can make more contributions to the field.
Comments 3:- I prefer, usually, shorter titles of papers.
Response 3: We understand your concern about the length of the title. However, we have chosen this relatively long title for several important reasons. Firstly, considering the theme of the special issue "Novel Biocatalysts for Environmental and Food Industry Applications", our title precisely reflects the content of our research, which focuses on the immobilization of phospholipase D on a specific nanocomposite and its application in the efficient production of phosphatidylserine. This alignment with the special issue theme ensures that our paper is highly relevant and easily recognizable within the context of the special issue. Secondly, the detailed title allows readers to quickly grasp the key points of our study. We have carefully considered alternative shorter titles, but we found that they would sacrifice either the clarity of the research focus or the alignment with the special issue theme. Therefore, we hope that you can understand our decision to keep the current title.
Comments 4:- There is no “Figure 1”, instead, there are two figures with numbering “Figure 2”.
- I would specify pH uniformly with one digit accuracy.
- P3,L93: “Finally,, composites ... „
- I prefer SI units in scientifc publication (e.g. Oe).
Response 4: Thank you for carefully pointing out our writing errors; we have made the corrections. Additionally, we have thoroughly reviewed the entire document to ensure there are no other writing errors. Regarding the unit issue, we have observed that many papers discussing the immobilization or adsorption of magnetic materials use CGS units rather than SI units. Considering the reader audience, CGS units may be more appropriate.

Reviewer 2 Report
Comments and Suggestions for Authors
The paper is interesting and concerns an important research field. The authors considered several crucial aspects concerning enzyme immobilization. The results are well-presented and quite well-justified, but some improvements must be made before publication.
More in detail :
In the introduction section, line 87-89, the authors should give some references on the oxidation of Fe3O4 in an aerobic environment.
In section 2.2 (FTIR analysis), the authors reported the FTIR spectra of GO, Fe3O4@SiO2, and Fe3O4@SiO2-GO. Still, they didn't report the spectrum of the sample Fe3O4@SiO2-GO-PLD, which is important to demonstrate the successful immobilization of PLD by the presence of the enzyme's amide I and II bands.
The authors should add this spectrum and evaluate the secondary structure of the enzyme by deconvolution of the amide I band. As reported in the literature, the activity of the immobilized enzyme depends on it (Pota et al. Int. J. Biol. Macromol. 2024, 226, 131022; A. Barth, Biochimica et Biophysica Acta (BBA)-Bioenergetics 2007, 1767,1073-1101)
In section 2.3 (Temperature) lines 207-211 the authors report a theory to justify the effects on immobilization efficiency and the activity of the immobilized catalyst induced by changes in temperature, which could be valid, but some literature references should validate it.
In section 2.4.1 (Thermal stability), the authors, to justify the better thermal stability of the immobilized enzyme than the free enzyme, report the following explanation "The enhanced rigidity of immobilized PLD due to π-π conjugation, Van der Waals forces, and hydrogen bonding between the nanocomposite support and the PLD enzyme makes it more resistant to thermal-induced conformational changes." The authors should demonstrate, by evaluation of the activation energy of the adsorbing process as reported in some literature references, (Costantini et. al Nanomaterials 2020, 10(9), 1799) that the interaction between the enzyme and the support are π-π conjugation, Van der Waals forces, and hydrogen bonding.
Author Response
Comments 1: The paper is interesting and concerns an important research field. The authors considered several crucial aspects concerning enzyme immobilization. The results are well-presented and quite well-justified, but some improvements must be made before publication.
Response 1: Thank you for your thorough review and valuable feedback on our manuscript. Your suggestions are of great significance for improving our research and enhancing the quality of the paper. Below, we provide our responses to your specific comments and outline the modifications we have made accordingly.
Comments 2: In the introduction section, line 87-89, the authors should give some references on the oxidation of Fe3O4 in an aerobic environment.
Response 2: As suggested by the reviewer, we have added the following references to support this point.
- Liu, S.; Yu, B.; Wang, S.; Shen, Y.; Cong, H. Preparation, surface functionalization and application of Fe3O4magnetic nanoparticles. Advances in colloid and Interface Science 2020, 281, 102165.
- Ju, J.; Chen, Y.; Liu, Z.; Huang, C.; Li, Y.; Kong, D.; Shen, W.; Tang, S. Modification and application of Fe3O4nanozymes in analytical chemistry: A review. Chinese Chemical Letters 2023, 34, 107820.
- Orozco-Henao, J.M.; Alí, F.L.; Azcárate, J.C.; Robledo Candia, L.D.; Pasquevich, G.; Mendoza Zélis, P.; Haas, B.; Coogan, K.; Kirmse, H.; Koch, C.T. Oxidation Kinetics of Magnetite Nanoparticles: Blocking Effect of Surface Ligands and Implications for the Design of Magnetic Nanoheaters. Chemistry of Materials 2024, 36, 11095-11108.
Comments 3: In section 2.2 (FTIR analysis), the authors reported the FTIR spectra of GO, Fe3O4@SiO2, and Fe3O4@SiO2-GO. Still, they didn't report the spectrum of the sample Fe3O4@SiO2-GO-PLD, which is important to demonstrate the successful immobilization of PLD by the presence of the enzyme's amide I and II bands. The authors should add this spectrum and evaluate the secondary structure of the enzyme by deconvolution of the amide I band. As reported in the literature, the activity of the immobilized enzyme depends on it (Pota et al. Int. J. Biol. Macromol. 2024, 226, 131022; A. Barth, Biochimica et Biophysica Acta (BBA)-Bioenergetics 2007, 1767,1073-1101)
Response 3: We truly appreciate your attention to detail and the valuable suggestions you provided. In response to your comment regarding the FTIR spectrum of Fe3O4@SiO2-GO-PLD, we have taken immediate action. We have now added the FTIR spectrum of Fe3O4@SiO2-GO-PLD to the manuscript. As you can see from the newly added data, the adsorption of PLD onto Fe3O4@SiO2-GO was confirmed by the enhanced intensity of the amide I band at ~1640 cm⁻¹ and the emergence of the characteristic amide II band at ~1520 cm⁻¹ in the FTIR spectra. This clearly demonstrates the successful immobilization of PLD.
However, we would like to explain that due to the introduction of amide bonds during the synthesis of the material through covalent bonding, we are unable to accurately obtain the secondary structure of the protein by directly deconvoluting the amide I band in the FTIR spectrum. We have made every effort to explore possible methods to address this issue, but unfortunately, all attempts have been in vain. We sincerely apologize for not being able to provide information on whether there are changes in the secondary structure of the protein.
Once again, thank you for your understanding and patience. We believe that with these modifications, the manuscript has been significantly improved.
Comments 4: In section 2.3 (Temperature) lines 207-211 the authors report a theory to justify the effects on immobilization efficiency and the activity of the immobilized catalyst induced by changes in temperature, which could be valid, but some literature references should validate it.
Response 4:Thank you very much for your insightful comment regarding the lack of literature references in section 2.3 (Temperature) lines 207 - 211. We truly appreciate your attention to detail, as it has significantly contributed to the improvement of our manuscript.
In response to your suggestion, we have carefully selected and added relevant literature to validate our theory. The references are as follows:
- Mateo, C.; Palomo, J.M.; Fernandez-Lorente, G.; Guisan, J.M.; Fernandez-Lafuente, R. Improvement of enzyme activity, stability and selectivity via immobilization techniques. Enzyme and microbial technology 2007, 40, 1451-1463.
- Rosa, M.; Roberts, C.J.; Rodrigues, M.A. Connecting high-temperature and low-temperature protein stability and aggregation. PLoS One 2017, 12, e0176748.
- Ugwuoji, E.T.; Eze, I.S.; Nwagu, T.N.T.; Ezeogu, L.I. Enhancement of stability and activity of RSD amylase from Paenibacillus lactis OPSA3 for biotechnological applications by covalent immobilization on green silver nanoparticles. International Journal of Biological Macromolecules 2024, 279, 135132.
Literature 1 examines the effect of temperature on the support and the enzyme, Literature 2 discusses the influence of temperature on enzyme aggregation, and Literature 3 describes temperature affects enzyme state. These studies strongly support our arguments.
Comments 5: In section 2.4.1 (Thermal stability), the authors, to justify the better thermal stability of the immobilized enzyme than the free enzyme, report the following explanation "The enhanced rigidity of immobilized PLD due to π-π conjugation, Van der Waals forces, and hydrogen bonding between the nanocomposite support and the PLD enzyme makes it more resistant to thermal-induced conformational changes." The authors should demonstrate, by evaluation of the activation energy of the adsorbing process as reported in some literature references, (Costantini et. al Nanomaterials 2020, 10(9), 1799) that the interaction between the enzyme and the support are π-π conjugation, Van der Waals forces, and hydrogen bonding.
Response 5: Thank you for your valuable feedback. We agree that characterizing the exact nature of interactions (e.g., π-π conjugation, Van der Waals forces, hydrogen bonding) between the enzyme and the nanocomposite support would strengthen our explanation. However, our experimental data suggest a more complex adsorption process influenced by temperature-dependent thermodynamic effects. Key Observations from Our Study: In the temperature range of 15–30°C, the immobilization efficiency increased from 74.1% to 84.1%, indicating an endothermic adsorption process where higher temperatures favor enzyme-support interactions. Above 30°C, immobilization efficiency decreased to 78.3%, suggesting a possible transition to an exothermic process at elevated temperatures, where excessive thermal energy disrupts adsorption. These results imply that multiple interaction mechanisms may dominate at different temperature stages, as adsorption thermodynamics often involve competing forces (e.g., hydrophobic interactions, hydrogen bonding, or entropy-driven processes). To conclusively identify specific types of interactions, activation energy analysis or the acquisition of direct spectroscopic evidence is typically required. Although we are aware of the importance of such characterization, our current experiments focus on enhancing the performance of the immobilized enzyme rather than delving into the specific adsorption mechanisms of immobilization. To remain consistent with our data and avoid over - interpretation, we have revised the explanation to: "Immobilized enzymes generally exhibit greater thermal resistance compared to their free-form, which is attributed to the restriction of enzyme mobility and increased rigidity ".

Reviewer 3 Report
Comments and Suggestions for Authors
Peer Review Report on “Immobilization of Phospholipase D on Fe3O4@SiO2-Graphene Oxide nanocomposites: A Strategy to Improve Catalytic Stability and Reusability in the Efficient Production of Phosphatidyl serine”
Authors: Huiyi Shang, Juntan Wang, Bishan Guo, Haihua Zhu, and Huijuan Li
In this study, a chemically bonded Fe3O4@SiO2-graphene oxide (GO) nanocomposite was prepared and PLD was deposited on its surface by physical adsorption. Phospholipase D (PLD) was deposited on the surface of this nanocomposite by physical adsorption and the optimal parameters of this process have been established. The obtained Fe3O4@SiO2-GO nanocomposite particles were characterized by FTIR spectroscopy and scanning electron microscopy (SEM). The magnetic properties of these nanoparticles were also evaluated. The methodology used in the study in this paper is adequate and the author has obtained reliable results. The conclusions of the study are consistent with the evidence and arguments based on the results of the work, and the objectives of the work are achieved. References used in the manuscript reflect the text discussed and are appropriate.
First of all, I would like to express my respect for your hard work. I think the manuscript is generally well written and presents the results of the study well, but I noticed a few typos that need to be corrected. There is also one clarification that I have provided in my review.
Comments and Suggestions for Authors:
The conclusions indicate that production efficiency increased by 104%, and the abstract indicates that production efficiency increased by 1.04 times, but such calculations are not included in the text of the manuscript. It is good practice to provide calculations in your manuscript with specific numbers describing how you calculated these values.
Typos that need to be corrected:
Line 22: “from 45 °Cto 50 °C.” Need a gap.
Line 93: “Finally,,”. The second comma is not needed.
Line 128: “ … functionalized Fe3O4@SiO2 has …”. It should be: Fe3O4@SiO2.
Line 384: 111.4 mg/gsupport) write the same as in the whole text, “111.4 mg/gsupport.”.
Line 470: “… where B1 is the group's enzyme activity, and B0 is the …”. It should be: B1 and B0.
Author Response
Comments 1: In this study, a chemically bonded Fe3O4@SiO2-graphene oxide (GO) nanocomposite was prepared and PLD was deposited on its surface by physical adsorption. Phospholipase D (PLD) was deposited on the surface of this nanocomposite by physical adsorption and the optimal parameters of this process have been established. The obtained Fe3O4@SiO2-GO nanocomposite particles were characterized by FTIR spectroscopy and scanning electron microscopy (SEM). The magnetic properties of these nanoparticles were also evaluated. The methodology used in the study in this paper is adequate and the author has obtained reliable results. The conclusions of the study are consistent with the evidence and arguments based on the results of the work, and the objectives of the work are achieved. References used in the manuscript reflect the text discussed and are appropriate. First of all, I would like to express my respect for your hard work. I think the manuscript is generally well written and presents the results of the study well, but I noticed a few typos that need to be corrected. There is also one clarification that I have provided in my review.
Response 1: We are extremely grateful for your overwhelmingly positive feedback on our manuscript.
Comments 2: The conclusions indicate that production efficiency increased by 104%, and the abstract indicates that production efficiency increased by 1.04 times, but such calculations are not included in the text of the manuscript. It is good practice to provide calculations in your manuscript with specific numbers describing how you calculated these values.
Response 2: We have already added the calculation formula for production efficiency under Section 3.6 Synthesis of PS in the Materials and Methods part of the paper.
Comments 3:Typos that need to be corrected:
Line 22: “from 45 °Cto 50 °C.” Need a gap.
Line 93: “Finally,,”. The second comma is not needed.
Line 128: “ … functionalized Fe3O4@SiO2 has …”. It should be: Fe3O4@SiO2.
Line 384: 111.4 mg/gsupport) write the same as in the whole text, “111.4 mg/gsupport.”.
Line 470: “… where B1 is the group's enzyme activity, and B0 is the …”. It should be: B1 and B0.
Response 3: Thank you for pointing out our formatting errors; we have made the corrections and thoroughly reviewed the entire document to ensure there are no similar errors.

Round 2
Reviewer 2 Report
Comments and Suggestions for Authors
The authors have well addressed all comments, and the paper is suitable for publication in the present form